# Changes in Resurgent Sodium Current Contribute to the Hyperexcitability of Muscles in Patients with Paramyotonia Congenita

**DOI:** 10.3390/biomedicines9010051

**Published:** 2021-01-08

**Authors:** Chiung-Wei Huang, Hsing-Jung Lai, Pi-Chen Lin, Ming-Jen Lee

**Affiliations:** 1Department of Post Baccalaureate Medicine, Kaohsiung Medical University, Kaohsiung 80708, Taiwan; g10054b@ms51.hinet.net; 2Institute of Physiology, Kaohsiung Medical University, Kaohsiung 80708, Taiwan; 3Department of Neurology, National Taiwan University Hospital, Taipei 10617, Taiwan; i5492111@gmail.com; 4Department of Neurology, National Taiwan University Hospital Jinshan Branch, New Taipei City 20844, Taiwan; 5Department of Internal Medicine, Division of Endocrinology and Metabolism, Kaohsiung Medical University Hospital, Kaohsiung 80756, Taiwan; pichli@kmu.edu.tw; 6Department of Neurology, National Taiwan University Hospital Yunlin Branch, Yunlin 640, Taiwan

**Keywords:** paramyotonia congenita, Na_v_1.4 channel, resurgent currents, sustained currents

## Abstract

Paramyotonia congenita (PMC) is a rare hereditary skeletal muscle disorder. The major symptom, muscle stiffness, is frequently induced by cold exposure and repetitive exercise. Mutations in human *SCN4A* gene, which encodes the α-subunit of Na_v_1.4 channel, are responsible for PMC. Mutation screening of *SCN4A* gene from two PMC families identified two missense mutations, p.T1313M and p.R1448H. To elucidate the electrophysiological abnormalities caused by the mutations, the p.T1313M, p.R1448H, and wild-type (WT) *SCN4A* genes were transient expressed on Chinese hamster ovary (CHO-K1) cells. The detailed study on the gating defects of the mutant channels using the whole-cell patch clamping technique was performed. The mutant Na_v_1.4 channels impaired the basic gating properties with increasing sustained and window currents during membrane depolarization and facilitated the genesis of resurgent currents during repolarization. The mutations caused a hyperpolarization shift in the fast inactivation and slightly enhanced the slow inactivation with an increase in half-maximal inactivation voltage. No differences were found in the decay kinetics of the tail current between mutant and WT channels. In addition to generating the larger resurgent sodium current, the time to peak in the mutant channels was longer than that in the WT channels. In conclusion, our results demonstrated that the mutations p.T1313M and p.R1448H in Na_v_1.4 channels can enhance fast inactivation, slow inactivation, and resurgent current, revealing that subtle changes in gating processes can influence the clinical phenotype.

## 1. Introduction

Myotonia is a condition characterized by hyperexcitability of muscle fibers after voluntary contraction or mechanical stimulation [1,2]. It manifests as spontaneous voluntary electrical activity of the skeletal muscle membrane, which is recorded as a “myotonic run” in electromyography (EMG) [1]. The voltage-gated Na^+^ channel (VGSC), Na_v_1.4, which is expressed on the skeletal muscle membrane, is a transmembrane complex. The human *SCN4A* gene, located on chromosome 17, encodes the α subunit of the Na_v_1.4 channel in skeletal muscles [3]. The Na_v_1.4 channel in the skeletal muscle is a heterodimer consisting of a pore-forming α-subunit and regulatory β1–4 subunits [4,5]. The Na_v_1.4 channel is composed of approximately 1836 amino acids, and the α-subunit consists of four homologous domains (DI–DIV), each containing six transmembrane segments. The VGSC is essential for the generation of action potentials at nerve tissues and also on muscle membranes. At depolarization, the S4 segments, which contain several positive amino acid residues and function as voltage sensors, can move outwardly and thereby alter channel confirmation to allow inward ion flux. The different charge residues in the S4 segments are domain-specific. The S4 of DI and DII are thought to play a prominent role in Na^+^ channel activation, while the S4 of DIII and DIV regulate fast inactivation [6]. The pore, with its selectivity filter for the inward ions, is lined by the loops, S5/S6, and individual S5 and S6 segments.

Depolarization of the Na_v_1.4 channel produces an action potential for muscle fiber contraction; however, continuous discharge is prevented by fast/slow inactivation, which guarantees normal contraction without myotonia [7,8,9]. Some mutations in the Na_v_1.4 channel result in alterations of muscle excitability [7,8]. Non-dystrophic myotonia with increasing excitability, such as paramyotonia congenita (PMC), is one of the clinical phenotypes [7,9]. The gain-of-function mechanism of PMC due to the mutation in Na_v_1.4 has been reported to enhance the inward Na^+^ currents, leading to the facilitation of activation and sometimes a hyperpolarized shift of the activation curves [10]. A few studies showed the slowed entry into fast inactivation and accelerated recovery from fast inactivation in the mutant channels from the PMC patients [11,12,13,14]. Furthermore, excitability is slightly enhanced during Na^+^ influx in PMC in cold environments. The biophysical dysfunction leads to an initial burst of myotonia discharges and results in muscle stiffness in PMC patients. Following activation, most Na_v_1.4 channels are inactivated and hold back the reactivation with an immediate successive stimulation, which may result in paralysis [7].

In two Taiwanese families with PMC, we found two known mutations, p.T1313M and p.R1448H [15], in the *SCN4A* gene. To investigate the mutant channels that induced the functional abnormality, whole-cell recording was employed to evaluate the changes in electrophysiology in Chinese hamster ovary (CHO-K1) cells expressing mutant *SCN4A* clones. The cellular study revealed that the p.T1313M and p.R1448H mutations altered the basic gating properties of the channel, increasing sustained and window currents at depolarization, and resurgent currents during repolarization. The increase of sustained and resurgent Na^+^ currents in mutant channels may be attributed to the destabilized inactivated states by acceleration of the transition from the inactivated Na_v_1.4 to the open states which induces hyperexcitability of muscle membrane. These findings not only suggest the molecular mechanism underlying the clinical presentations of PMC, but also strongly implicate that p.T1313 (DIII-IV linker) and p.R1448 (DIV/S4) play critical roles in the molecular operations of recovery from fast inactivation and resurgent Na^+^ current genesis.

## 2. Experimental Section

### 2.1. Patients and Genetic Analysis

Two index patients developed muscle stiffness with difficulties of relaxation and hypertrophic muscle bulk since childhood. Muscle contraction became more severe, and transient weakness with some aching pain occurred after repetitive exercise and cold exposure. There was no muscle wasting or sensory deficits. The clinical impression was paramyotonia congenital (PMC), which is mainly caused by the mutations in human *SCN4A* gene. Genetic analysis from the two patients was carried out by Sanger sequencing of the candidate gene. All the genetic analysis protocols were approved by the Research Ethics Committee of the National Taiwan University Hospital (201802049RINB, 23/4/2018-22/4/2019), Taipei, Taiwan.

### 2.2. Preparation of cDNA Constructs for Mutagenesis

The cDNA clone (pcDNA3.1(+)-DKY) of the wild-type (WT) alpha subunit of human voltage-gated Na^+^ channel, Na_v_1.4 (*SCN4A*), was obtained from OriGene Technologies Company (Cat. No. RC218290; Rockville, MD, USA) [16,17]. The cDNA for the p.T1313M and p.R1488H mutant channels were made using a QuikChange^®^ site-directed mutagenesis system kit (Stratagene, La Jolla, CA, USA). The variants at c.3938C>T (p.T1313M) and c.4343G>A (p.R1448H) were replaced in the cDNA clone. The sequences were confirmed by automatic DNA sequencing (3730xl DNA Analyzer; Applied Biosystems, Foster, CA, USA).

### 2.3. Cell Cultures and cDNA Transfection

Chinese hamster ovary-K1 (CHO-K1) cells were purchased from the Food Industry Research and Development Institute, Hsinchu, Taiwan. The manipulation of CHO-K1 cells conformed to the ethical information guidelines of the Kaohsiung Medical University, and the protocols were approved by the Institutional Biosafety and Use Committee. The CHO-K1 cells were incubated in F12-K culture medium (Thermo Fisher Scientific, Waltham, MA, USA) under humidified conditions at 37 °C with 95% O_2_/5% CO_2_. The F12-K culture medium was supplemented with 10% fetal bovine serum (Thermo Fisher Scientific) and 0.5% penicillin–streptomycin–ampicillin solution (Thermo Fisher Scientific). CHO-K1 cells (1 × 10^6^ cells) were seeded onto a 35-mm cell culture dish (Greenpia Technology, Seoul, South Korea) and transfected with cDNA clones of WT, p.T1313M and p.R1488H mutant Na_v_1.4 channels using Lipofectamine™ 3000 (Thermo Fisher Scientific). The 5.0 μg cDNA construct was also mixed with 0.1 μg green fluorescent protein. All reagents were added in the F12-K medium and incubated for approximately 4 days.

### 2.4. Electrophysiological Recordings

Prior to taking electrophysiological recordings, proteinases II–XIII (0.1 mg/mL; Sigma-Aldrich, St. Louis, MO, USA) were added to the transfected cells. The transfected cells were then plated onto coverslips at 37 °C for approximately 60 min. Electrophysiological recordings were carried out within 4 days after transfection. Currents were recorded at approximately 25 °C with an Axopatch 700B amplifier (Axon Instruments, Sunnyvale, CA, USA) interfaced with pClamp 9.0 acquisition software (Molecular Devices, San Jose, CA, USA). Currents were filtered at 5 kHz with a four-pore Bessel filter and digitized at 50 μs intervals using the Digidata-1322A interface, a signal conditioning amplifier (Axon Instruments, Union City, CA, USA). The electrophysiological recordings were obtained using fire-polished, borosilicate glass-pulled micropipettes with a tip diameter of ~1.0 μm (pipette resistance was 1.5–2.5 mΩ). Micropipettes were prepared using the Sutter P-97 puller (Sutter Instrument Company, Novato, CA, USA). The glass electrode pipette was filled with an internal solution containing 75 mM CsCl, 75 mM CsF, 5 mM HEPES, 2 mM CaCl_2_, and 2.5 mM EGTA (pH 7.4 to 7.6). The whole-cell configuration was immersed in an external solution containing 145 mM NaCl, 10 mM HEPES, 2 mM CaCl_2_, and 2.5 mM MgCl_2_ (pH 7.4 to 7.6). An intracellular Na_v_β4 peptide (KKLITFILKKTREK-OH, 100 μM, Genomics Bioscience and Technology Co., Ltd, Taiwan) was also added to the intracellular solution to generate the resurgent currents. Tetrodotoxin (TTX) (1.0 μM) (Tocris, Bristol, UK) was used to inhibit TTX-sensitive (TTX-s) Na^+^ currents. Resurgent currents were obtained by subtracting the TTX-s Na^+^ currents.

### 2.5. Homology Modeling

The homology modeling procedure was performed using methods similar to those previously used [18,19,20]. A homology model of the WT Na_v_1.4 channel encoded by the *SCN4A* gene was built from X-ray crystal structure data corresponding to the human voltage-gated Na_v_1.4 channel (human Na_v_1.4; PDB code: 6AGF) [21]. The amino acid sequence of the WT human Na_v_1.4 channel was obtained from the UniProt database (P35499). The aligned sequences of the WT Na_v_1.4, p.T1313M, and p.R1488H mutant channels were processed using Discovery Studio 2018 (Dassault Systèmes, Vélizy-Villacoublay, France) to generate the secondary structures and assign the relative positions [18,19,20].

### 2.6. Measurement of the Fast Steady-State of Activation and Inactivation Curves

For the current–voltage (I–V) plot, patch cell configurations were held at −120 mV and subjected to different test pulses increasing in 5-mV increments from −160 to +40 mV for approximately 100 ms [18,19,20]. The peak amplitude of inward Na^+^ currents was plotted against test membrane potentials to give a I–V plot, including a progressive line between 0 and +40 mV. The reversal potential of Na^+^ ions was determined by the straight line intersecting the transverse axis (V_m_). The maximal Na^+^ conductance (G_max_) was given by the slope of this regressive line (G/G_max_). The normalized Na^+^ conductance was defined as I_peak_/[(V − V_Na_^+^) × G_max_], where I_peak_ and V show the peak Na^+^ currents and test potentials during different membrane depolarizations, respectively. The normalized Na^+^ conductance was plotted against the membrane potential, and all plots were fitted using Boltzmann function to make the activation curve: G/G_max_ = 1/[1 + exp(V_h_ − V)/k)], where k is the slope factor, G_max_ is maximal Na^+^ conductance, V_h_ is the membrane potential at half activation, and V is the membrane potential. For the steady-state inactivation curve of Na^+^ currents, the maximal current obtained with a +10-mV test pulse was documented after a 100-ms prepulse at various voltages from a holding potential of −120 mV. The maximal amplitudes obtained at the test pulse were normalized to the maximal amplitude (I/I_max_) and plotted against the prepulse membrane potentials to create the inactivation curve. The inactivation curve was fitted with Boltzmann function: I/I_max_ = 1/[1 + exp(V − V_h_)/*k*)], where I_max_ is the maximal amplitude, k is the slope factor, V_h_ is the potential at half inactivation, and V is the prepulse membrane potential. SigmaPlot 10.0 software (Systat Software, San Jose, CA, USA) was used to depict the fit curves according to the sets of activation and inactivation experimental data [18,19,20].

### 2.7. Data Analysis

All statistical data were described as mean ± standard error mean. The data were assessed and analyzed using Student’s independent *t*-test(s), and statistical significance was denoted by *p* < 0.05.

## 3. Results

### 3.1. The Sustained Na^+^ Currents in the p.T1313M and p.R1448H Mutant Channels Were Larger than Those in the WT Na_v_1.4 Channel

Two index patients developed muscle stiffness with difficulties in muscle relaxation. The symptoms aggravated after repetitive exercise and cold exposure. Two sequence variants c.3938C>T (p.T1313M) and c.4343G>A (p.R1448H) were identified from the patients, which fulfilled the diagnosis of paramyotonia congenita (PMC). To elucidate the functional disturbances caused by the mutations, patch-clamp technique for whole cell recording on the transient expressed CHO-K1 cells was carried out. The comparison of the electrophysiological properties of the gating control among the three Na_v_1.4 channels (WT, p.T1313M, and p.R1448H) is shown in Figure 1A and Table 1. In comparison with the WT channel, the activation curves displayed a hyperpolarization shift in the p.R1448H, but not the p.T1313M, mutant channel (Figure 1B and Table 1). The inactivation curves moved toward hyperpolarization in the p.T1313M and p.R1448H mutant channels (Figure 1B and Table 1). The slopes of the activation and inactivation curves of the mutant channels did not exhibit any significant changes compared with the WT Na_v_1.4 channel (Figure 1B and Table 1). The predicted window current or the presumed sustained Na^+^ current was defined as the measurement of area under the activation and inactivation curves. As shown in Figure 2A,B, the window currents in the p.T1313M and p.R1448H mutant channels were significantly larger than those in the WT Na_v_1.4 channel. The productions of the relative conductance (G/G_max_) and steady-state inactivation (I/I_max_) at activation and inactivation curves at −20, −40, −60, and −80 mV (Figure 2C) were used to represent the window or sustained Na^+^ currents at the specific potentials. Production in the p.T1313M and p.R1448H mutant channels was significantly larger than that in the WT Na_v_1.4 channel (Figure 2C). Likewise, when the ratio of sustained current to peak current was compared among the three channels, the ratio for the p.T1313M and p.R1448H mutant channels was also significantly larger than that for the WT Na_v_1.4 channel (Figure 2D and Appendix A). The sustained Na^+^ currents in the p.T1313M and p.R1448H mutant channels increased significantly, indicating that the p.T1313M and p.R1448H mutant channels obtained larger currents during depolarization (Figure 2D and Appendix A). Although there was some hyperpolarizing shift in the mutant channel at activation, the hyperpolarizing shift during fast inactivation compensated the effect, resulting in an increase in window or sustained Na^+^ current in the mutant channels.

### 3.2. The Maximal Resurgent Na^+^ Currents in the p.T1313M and p.R1448H Mutant Channels Were between −80 and −10 mV

Previous studies showed that resurgent Na^+^ currents can be readily detected in the presence of intracellular Na_v_β4 peptides [18,19,20]. We then investigated whether the mutations in Na_v_1.4 channel can cause any biophysical changes of the resurgent Na^+^ current. Following a depolarization prepulse at +40 mV, the resurgent current in the WT, p.T1313M, and p.R1448H mutant channels was generated at the different voltages of repolarization, ranging from 0 to −100 mV (Figure 3). The resurgent Na^+^ currents in the mutant channels were significantly larger than those in the WT Na_v_1.4 channel (Figure 3B). The p.T1313M mutation is located in the linker between the DIII and DIV domains, whereas p.R1448H is located at *DIV/S4*. Both mutations generated larger resurgent Na^+^ currents than the WT channel.

### 3.3. The WT Na_v_1.4 Channel Exhibits Two Different Activation Curves for Transient Na^+^ and Resurgent Na^+^ Currents

We examined the voltage-dependent occupation of the open state, which can induce resurgent Na^+^ currents. A depolarization with a duration of 10 ms induced a similar activation curve for resurgent Na^+^ currents in the WT, p.T1313M, and p.R1448H mutant channels (Figure 4A,B). With a 10-ms prepulse of depolarization, most resurgent Na^+^ channels will enter the open state. Comparison of the activation curves between the resurgent and transient Na^+^ currents revealed that the activation curves of resurgent Na^+^ currents had decreased voltage dependence. When compared with the activating curves of transient Na^+^ current, there was an apparent depolarizing shift toward the positive voltages in those of resurgent current (Figure 4B). In addition to the significant increase in sustained and resurgent Na^+^ currents of the p.T1313M and p.R1448H mutant channels (Figure 2D and Figure 3B), the resurgent Na^+^ currents in the mutant channels showed somewhat hyperpolarizing shift as compared to that of the WT, especially at a low repolarization voltage (Figure 4B). These findings may suggest that the mutations have a lower binding tendency of the inactivation particle to the available open Na^+^ channel, which results in increases in the resurgent and sustained Na^+^ currents.

### 3.4. Delayed Times to Peak of Resurgent Na^+^ Currents in the p.T1313M and p.R1448H Mutant Channels than in the WT Na_v_1.4 Channel

Given the increase in resurgent Na^+^ currents in the mutant channels, we evaluated the changes of sodium current kinetics in the p.T1313M and p.R1448H mutant channels (Figure 5A,B). When the depolarizing prepulses were increased from +40 to +180 mV, the decay kinetics (1/tau) of the resurgent Na^+^ currents did not change significantly, as compared to the WT channel (Figure 5A). However, the time to peak for the resurgent Na^+^ currents in the mutants was significantly longer than that in the WT (Figure 5B). In addition to the different depolarizing prepulses test, we also evaluated the kinetics of resurgent Na^+^ currents with different repolarizing voltages from −70 to −20 mV in the mutant channels. As shown in Figure 5C, both mutant channels showed significantly longer times to reach the peak, as compared to the WT channel. The time was more protracted in the mutant channels with a higher repolarizing voltage (−20 mV) than that found with a low voltage (−70 mV) (Figure 5C). The decay kinetics of the resurgent Na^+^ currents followed the membrane hyperpolarization. The decay rates from the open states of the p.T1313M and p.R1448H channels, which generate the resurgent Na^+^ currents, mimicked those of the WT channel (Figure 5D). Since the tail current also occurs after depolarization, the decay kinetics of the tail current between the mutant and WT Na_v_1.4 channels were also assessed. With a longer repolarization, we found that the deactivation rates in both WT and mutant channels were less than 0.4 ms^−^^1^ (Figure 5E,F). Interestingly, the inverses of the decay time constants of the resurgent currents were five- to tenfold slower than the deactivating times of the tail currents (Figure 5D–F). These findings indicate that the Na_v_β4 peptide affected the deactivation rate from the open to the closed state and there might be the existence of two different open states responsible for the transient, and resurgent Na^+^ currents, respectively.

### 3.5. Slow Inactivation in WT Na_v_1.4, p.T1313M, and p.R1448H Mutant Channels

In addition to fast inactivation, Na_v_1.4 channels may enter into a slow inactivated state after depolarization, which develops on a time scale of a few seconds [22,23,24]. We measured the voltage dependence of slow inactivation using an 18-s conditioning pulse. We also introduced an intermediate 0.5-s hyperpolarized pulse to allow recovery from fast inactivation before assessing the availability of the channels to open (Figure 6A). The maximal Na^+^ current amplitude measured during the test pulse was normalized and reported as a function of the 18-s conditioning pulse voltage. The relationships were fitted with a Boltzmann function containing residual currents because ~30% of channels did not inactivate at a positive voltage in a short time period (0 mV; 0.2 s) (Figure 6B). The p.T1313M and p.R1448H mutant channels significantly affected the slow inactivation curve and the half-maximal inactivation voltage (V_1/2_); however, they did not reduce the maximal number of inactivating channels (Figure 6C,D). The voltage to attain half of the slow inactivation was much lower in the mutant channels than in the WT channel.

### 3.6. Reduced Magnitudes of the Resurgent Currents by Prolonging of the Depolarization Prepulses

To further investigate the kinetics of mutant resurgent currents, we examined the changes in the resurgent currents in both WT and mutant channels after gradually lengthening the depolarizing prepulse (Figure 7A). Protracting the time of depolarization, the resurgent current decreased in all the tested channels (Figure 7A,B). These findings indicate that a ~5-ms depolarization prepulse does not allow the WT, p.T1313M, or p.R1448H Na_v_1.4 channels to reach a steady-state distribution of inactivated channels. While it is easy to reach the new “resurgent” open and corresponding inactivated states in ~5 ms, the subsequent distribution of the channel protein favors the conventional inactivated state, which does not require an open state to be deactivated during the subsequent repolarization. The voltage-dependent kinetics and the “steady-state” relative residual resurgent currents were similar among the WT, p.T1313M, and p.R1448H Na_v_1.4 channels. However, the absolute speed of decay was ~threefold slower in the p.T1313M and p.R1448H channels than in the WT Na_v_1.4 channel (Figure 7C,D).

### 3.7. Topological Changes in the p.T1313M and p.R1448H Mutant Channels Were Shown by Homology Modeling

In the WT Na_v_1.4 channel, p.T1313 is located at the linker between the DIII and DIV domains, and p.R1448 is located at DIV/S4. We checked the topological changes caused by the missense mutations. The benzene ring with hydroxyl group of tyrosine is replaced by the nonpolar dimethylsulfite side chain of methionine in the p.T1313M mutation. The p.T1313 is located next to the “IFM” (amino acids at 1310-1311-1312) motif, which is critical for channel inactivation. Homology modeling of the Na_v_1.4 channel (Figure 8) showed that the p.T1313M mutation may increase the distance between p.T1313 and the nearby residues in domain III (e.g., p.L1482, p.E1599, and p.N1602; Figure 8C,D).

The α-helix, DIV/S4 harbors a few positive residues as the voltage sensor. At depolarization, these positive residues move outwardly and open the channel. The residue, p.R1448 is located close to the outer membrane and plays a role as a voltage sensor. The missense mutation, p.R1448H replaced the long side chain of arginine by the aromatic ring of histidine. The mutation channel enlarges the distance between p.R1448 and residues located in domain IV (e.g., p.D1383, p.L1385, and p.I1387). The structural changes resulting from the mutation may be a critical factor in channel activation/inactivation, which may lead to weak inactivation and uncoupling between activation and inactivation. The structural changes related to the p.T1313M and p.R1448H channels may be a critical factor in channel activation and inactivation, which may result in weakened inactivation and uncoupling between activation and inactivation.

## 4. Discussion

### 4.1. Sodium Channels Are Associated with Excitation-Contraction Coupling in the Skeletal Muscle

Compared with the neuronal resting membrane potential (V_m_ = ~−65 mV), the skeletal muscle maintains a more negative resting membrane potential, usually closer to −90 mV, since there is an increased gradient of K^+^ and Cl^−^ ions with the greater resting chloride ion permeability. Along with the inward rectifier potassium channel (Kir2.1), the transverse tubule (T-tubule) membrane contains a chloride channel (ClC-1) that contributes to the V_m_. The VGSC and its associated proteins are expressed on the plasma membrane of the T-tubules and the sarcolemma of skeletal muscles [25,26]. The sodium channel density is approximately 100-fold higher at the endplate of the neuromuscular junction (NMJ) [27], which contributes to the high safety factor of synaptic transmission and causes a muscle fiber action potential to be elicited from each motor-neuron action potential. The motor-neuron releases acetylcholine (ACh) into the NMJ, which opens the ACh-gated Na^+^ channels to depolarize the muscle membrane. Based on such depolarization, the VGSCs activate rapidly (<1 ms) and induce a large inward sodium current (~5 mA/cm^2^) to drive a fast stroke of action potential (dV/dt = ~500 mV/ms). The action potential on the muscle membrane propagates into the T-tubules, triggering a conformational change in the voltage-sensitive dihydropyridine (DHP) receptor to initiate excitation–contraction (E-C) coupling [28,29]. The receptor primarily acts as a voltage-sensitive protein in the skeletal muscle that changes conformation with depolarization. The DHP receptor physically interacts with ryanodine receptor/calcium release channels in the sarcoplasmic reticulum membrane and allows calcium to flow into the cytoplasm to induce muscle contraction. Accordingly, the VGSC Na_v_1.4 channel, which is mainly expressed on the muscle membrane, plays a key role in inciting E-C coupling.

For the functional study of the mutant Na_v_1.4 channel, heterologous expression of the mutant clones in mammalian cell lines (HEK-293T or tsA201 cells) or *Xenopus* oocytes has been employed [30]. In previous studies, the Na_v_1.4 mutations, such as p.G270L (located in DI/S5–6 linker), p.N440K (located in DI/S6), p.G1306E (located in DIII–IV linker), p.T1313A (located in DIII–IV linker), N1366S (located in DIV/S1), and p.R1448P (located in DIV/S4), that cause PMC were shown to cause a depolarizing shift in the inactivation curve and/or to slow the macroscopic inactivation kinetics of the channel [12,31,32,33]. Other mutations associated with PMC, such as p.I693L (located in DII/S2), p.G1456E (located in DIV/S4), p.R1448C (located in DIV/S4), p.R1448H (located in DIV/S4), and p.A1589D (located in DIV/S5), have been found to be associated with muscle channelopathy to cause the myotonia [8,11,12,14,31,34,35,36,37,38]. However, the molecular mechanism involved remains elusive. More recently, we identified a missense mutation, p.V445M, in the Na_v_1.4 channel from two families with nondystrophic myotonia [20]. The electrophysiological study to evaluate the changes in transient and resurgent currents in WT and mutant channels revealed that both sustained and resurgent Na^+^ currents were increased in the mutant channels. With the expression of small Na_v_β4 peptides, the resurgent Na^+^ current occurred at the repolarization stage with reopening of the VGSC. Therefore, the resurgent current further reduced the threshold for the generation of successive firing. It has been proposed that, during the inactivating stage, there is competition between the Na_v_β4 peptides and the inactivating molecules to block the reopened channel [39,40,41,42].

The mutation, p.T1313M replaces the side chain of the polar uncharged group of threonine by a nonpolar aliphatic dimethylsulfite group of methionine. The p.T1313 is next to the IFM motif which plays a critical role for inactivation. The high resolution of three-dimensional (3D) structure of Na_v_1.4 channel with β1 subunit showed that the Thr^1313^ is adjacent to the motif for fast inactivation, IFM [6]. The polar interaction of Thr^1313^-Gln^1316^ and Thr^1313^-Asn^1484^ may have an impact on the allosteric docking mechanism for IFM motif. The distance for Thr^1313^-Gln^1316^ and Thr^1313^-Asn^1484^ are 4.35, and 7.90. When the Thr^1313^ replaced by methionine, the distance becomes 4.25 and 7.45, respectively. As shown in Figure 1B, the inactivation curve of the p.T1313M mutant channel reveals a hyperpolarizing shift relative to the WT channel. The mutation may cause the aberrant allosteric docking of the IFM motif leading to the change in fast inactivation.

The amino acid, p.R1448 is at DIV/S4 which plays a role as voltage sensor implicating the activation of the voltage gated channel. According to the homology modelling, p.R1448 is located close to the outer membrane. The change of arginine to histidine replaced the long-branch side chain by the aromatic ring, although both of the side chains are positively charged. The biophysical defects caused by the mutation, p.R1135H in Na_v_1.4 results in the phenotype of hypokalemic periodic paralysis [43]. The Arg1135 is located at DIII/S4 and also plays a role as voltage sensor. The molecular dynamic simulations suggested that the p.R1135H substitution causes the disruption interactions of DII/S2 countercharges with the DIII/S4 during repolarization of the membrane. The p.R1448H mutation causes the hyperpolarizing shift of the activation curve of the mutant channel (Figure 1B), which might be related to the aberrant interactions of the countercharges in S1~S3. As we envisaged that p.T1313M and p.R1448H mutations in Na_v_1.4 channel lead to the changes in inactivation and activation processes, respectively. In terms of the impact of these mutations on resurgent current, this study showed that the increase in resurgent current and prolonged the time to peak in the mutant channels. The increased distance between the mutant residues and the surrounding amino acids implies the conformation change, which may contribute to the disturbances of gating properties.

### 4.2. Molecular and Biophysical Basis of Resurgent Na^+^ Currents

Resurgent Na^+^ current is defined as the ion current through sodium channels which reopen in response to negative voltage changes after the inactivation of macroscopic transient current [39]. The resurgent Na^+^ current may readily reduce the threshold to fire another action potential [40,44,45,46,47], leading to rapidly repetitive discharges. This hyperexcitability is essential in several physiological and pathophysiological phenomena, such as myotonia, erythromelalgia, and paroxysmal extreme pain disorder [40,46,47]. Resurgent Na^+^ currents occur in cerebellar, vestibular, and subthalamic neurons, which are capable of firing in bursts, even in response to a single stimulus [40,45,46,47]. The expression of specific VGSC types, such as Na_v_1.4 and Na_v_1.6, has been reported to be capable of generating the resurgent currents [48]. The relatively slow inactivation speed of the Na_v_1.4 channel may also be a key factor that contributes to the genesis of resurgent currents. How the mutations cause the molecular dynamic changes between these two routes of recovery from the inactivated VGSC remains unclear. The “ball-and-chain” model explains the classical mechanistic view of the fast inactivation after an opened state of the channel [49]. In this model, fast inactivation results from blockage of the activated channel pore by part of the channel protein (the “ball” or the “inactivating particle”), and repolarization occurs after the channel is inactivated.

To circumvent the established observation that the closed inactivation door can only be opened after the activation gate is closed [50], Grieco et al., (2005) proposed that the genesis of resurgent currents depending on the presence of a “resurgent particle” [51]. The resurgent particle is thought to compete with the inactivating particle to block the open channel pore, but does not allow the activation gate to close (deactivate) until the resurgent particle has left its blocking site. Grieco et al. also suggested that the cytoplasmic sequences of the β4 subunits, which contain several positive charges and clustered hydrophobic/aromatic residues, may block the pore and act as resurgent particles. However, this model does not explain some of the observations in this and our previous studies [20]. Firstly, the size of the expressed Na_v_β4 subunit is 14 amino acid peptides, which is much smaller than the inactivating particle. The time constant for the decay of resurgent current is around a few tens of milliseconds rather than only a few milliseconds that we found in fast inactivation [44,46,51]. Secondly, the slope for the activation curves of the transient and resurgent Na^+^ currents is different. Furthermore, there shall be a ‘hyperpolarizing’ shift of the activation curve of the resurgent current compared to that of the transient current according to the competitive blocking model. On the contrary, the activation curve of resurgent current from the Na_v_1.4 channel shifts to a more positive potential (a ‘depolarizing’ shift) (Figure 5B). These findings from the Na_v_1.4 channel are in line with our previous study on the Na_v_1.7 channel. Both are difficult to envisage using the conventional model [18,19,20], due to (1) the concomitant increase in resurgent and sustained currents, (2) the slow decay phase of transient currents only in the presence of the Na_v_β4 peptide, and (3) the lengthened time to peak of the decay kinetics and the activation curve of resurgent currents. Therefore, as we proposed earlier, there may be at least two distinct open states of the resurgent current generated by VGSC, which are responsible for transient and resurgent currents, respectively (Figure 9B in ref. [18]). According to the new scheme, the Na_v_β4 peptide is mainly a modifier that induces the new gating conformations, rather than acting as a pore blocker that competes with the inactivating particle [18,19,20]. The generation of resurgent currents may be due to a global gating conformational change. Furthermore, mutations in the DIII–IV linker (p.T1313M) and D4/S4 (p.R1448H) in the Na_v_1.4 channel are prone to generating resurgent current and have a delay time to peak for decay of resurgent current, resulting in hyperexcitability with repetitive discharge.

To elucidate the impact of mutations of *SCN4A* gene on the channel function, the changes of transient activation and inactivation curves of the mutant channels have been studied. However, the shifting of activation and inactivation curves of the channels with p.T1313M and p.R1448H mutations relative to WT channel are not consistent, although the impaired basic gating properties, leading to the increase of sustained and window currents during membrane depolarization.

While investigating the resurgent sodium current from the mutant channels, both the mutations results in increasing the I/I_max_ of the resurgent current and in the prolonged the time to peak. These findings are in line with our previous study on the p.V445M mutant channel [19]. The generation of large resurgent currents as well as the prolonged time to peak can be replicated among these mutant channels. Therefore, we argue that the changes in resurgent current may play an important role to cause the hyperexcitability of the affected muscles, although other factors such as the shift of the transient activation and inactivation curves, and the increase of sustained and window currents may also contribute to the phenotype.

In some mutations in Na_v_1.4 that cause myotonia, defects in slow inactivation have been reported [52,53]. In line with these studies, the V_1/2_ of slow inactivation in the mutant channels significantly increased, but the maximal number of inactivating channels was not reduced. These findings suggest the enhancement of slow inactivation, which develops faster in the mutant channels than in the WT channels. In addition to the enhanced slow inactivation, analyzing the decay kinetics of the tail currents from the Na_v_1.4 channel with the p.T1313A mutation was only slow at positive potentials [12]. As shown in Figure 5E,F, there was no significant change in time constant for the decay kinetics of tail current in the mutant channel, as compared to the WT channel. This finding may be due to the different amino acid substitution; the side chain methionine (p.T1313M) is much larger than alanine (p.T1313A), although both missense mutations occur in PMC patients. The different expression system may also be a factor for such a discrepancy. The decay kinetics showed that the deactivation rate is around 0.4 ms^−^^1^. However, it is five- to tenfold slower than that of the resurgent current. These findings also confirm that the resurgent current is independent of slow inactivation and the tail current.

### 4.3. Factors Contributed to the Hyperexcitability of Affected Muscles

The cooling effect on the Na_v_1.4 channel with the missense mutations, p.T1313A and p.R1448H has been explored previously [12,13,38]. In the absence of Na_v_1.4 peptides, the temperature sensitivity of the Na_v_1.4 channel is not modified by the mutation p.T1313A; but, the gating kinetics were slowed in mutant channel upon cooling exposure, which reached to the threshold for myotonia [12]. At lower temperature, the voltage-dependent activation curve of the p.R1448H was shift to more negative potentials than that found in WT channel [38]. Moreover, the window current was increased in the mutant channels at cooling temperatures. These findings demonstrate that the influence of lowing of temperature can enhance the biophysical properties of the transient sodium current conferring to hyperexcitability of muscle fibers.

While considering the temperature effect on the resurgent current, we had investigated the changes in electrophysiological properties on Na_v_1.7 channels with different mutations, p.I136V, p. I848T, and p.V1316A. These mutations were identified from patients with erythromelalgia (IEM), which was provoked by warm temperature or fever. The influence of temperature on the gating properties and electrophysiological manifestations had been addressed [19]. The in vitro biophysical study of the mutant channels showed a consistent temperature-dependent enhancement of the relative resurgent currents normalized to the transient currents. In addition, we found that there is a temperature-dependent change in the time to peak and in the decay kinetics of the resurgent currents of the Na_v_1.7 channel. The changes of electrophysiological properties of the resurgent current in Na_v_1.4 channel mediated by the cooling temperatures need to be elucidated in future.

### 4.4. Alteration in Fast Inactivation and Increase of Persistent Sodium Current in p.T1313 and p.R1448 Mutant Channels Causing for Cellular Hyperexcitability

In previous study, the slow inactivation did not show any disparate results in p.T1313M and p.R1448C mutant channel compared to WT channel, although a significant alteration in fast inactivation was identified [14]. The fast inactivation including the slowing down of the open-state inactivation was envisaged as the cause for cellular hyperexcitability [14]. The present study showed the hyperpolarization shifting in voltage-dependent availability of open state sodium channel from inactivation [11]. Like p.T1313M, the Na_v_1.4 channel with p.T1313A mutation also shows the impediment of Na^+^ channel fast inactivation, despite the slowing and reducing the voltage sensitivity of kinetics and decreasing the voltage-dependence of steady state [12,13]. Moreover, by using slow depolarizing ramps meant that the presence of increase persistent sodium current (I_Na_) could be observed in the cells expressing the mutation of p.R1448C [33]. In summary, these studies demonstrated that mutations at the p.T1313 and p.R1448 cause an impact on the fast inactivation and increase of persistent sodium current leading to hyperexcitability of muscle membrane.

## 5. Conclusions

Mutations of the Na_v_1.4 channel are responsible for myotonic myopathy, including myotonic congenita, PMC, and periodic paralysis. In addition, some patients with painful contraction have also been reported [30]. More recently, sudden infant death syndrome has been found to be correlated with mutations in the Na_v_1.4 channel [54]. Despite the variability of clinical features caused by *SCN4A* mutations, the biological dysfunctions caused by these mutations need to be investigated. The biophysical functional defects in transient and resurgent currents from the Na_v_1.4 channel were evaluated by whole-cell recording. The transient sodium current evaluation revealed an increase in sustained and window currents in the mutant channels. During the repolarizing stage, the recording revealed a higher resurgent current with a slow time to peak in the mutant channels as compared to the WT channels. In addition to the changes in fast- and slow- inactivation of transient current, the increase of resurgent current with slow time to peak in the mutant channel confers the hyperexcitability of muscle membrane with repetitive firing leading to the cardinal feature, myotonia.

## Figures and Tables

**Figure 1 biomedicines-09-00051-f001:**
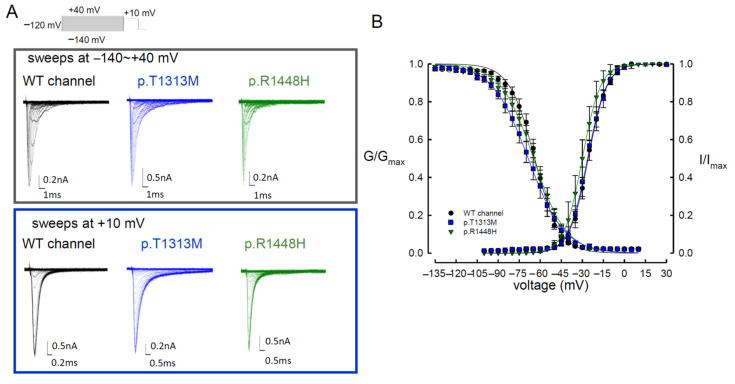
Activation and Inactivation Curves of WT, p.T1313M, and p.R1448H Mutant Na_v_1.4 Channels in the Presence of 100 μM Na_v_β4 Peptide. (**A**) Representative current traces for the WT, p.T1313M, and p.R1448H mutant Na_v_1.4 channels (activation in the upper panel, inactivation in the lower panel) in the presence of 100-μM Na_v_β4 peptide. (**B**) The steady states of the activation and fast inactivation curves of the WT, p.T1313M, and p.R1448H mutant Na_v_1.4 channels were fitted with Boltzmann function: 1/[1 + exp((V_h_ − V)/k)], where V is the membrane voltage, V_h_ and k are −24.7 ± 1.5 mV and 6.85 ± 0.25 for the activation curve and −61.9 ± 0.9 mV and −8.9 ± 0.3 for the fast inactivation curve in the WT Na_v_1.4 channel, −26.29 ± 1.8 mV and 6.76 ± 0.57 for the activation curve and −68.6 ± 0.34 mV and −12.82 ± 0.28 for the fast inactivation curve for the p.T1313M mutant channel, and −30.0 ± 1.5 mV and 6.4 ± 0.45 for the activation curve and −64.78 ± 0.45 mV and −11.6 ± 0.38 for the fast inactivation curve for the p.R1448H mutant channel (*n* = 5).

**Figure 2 biomedicines-09-00051-f002:**
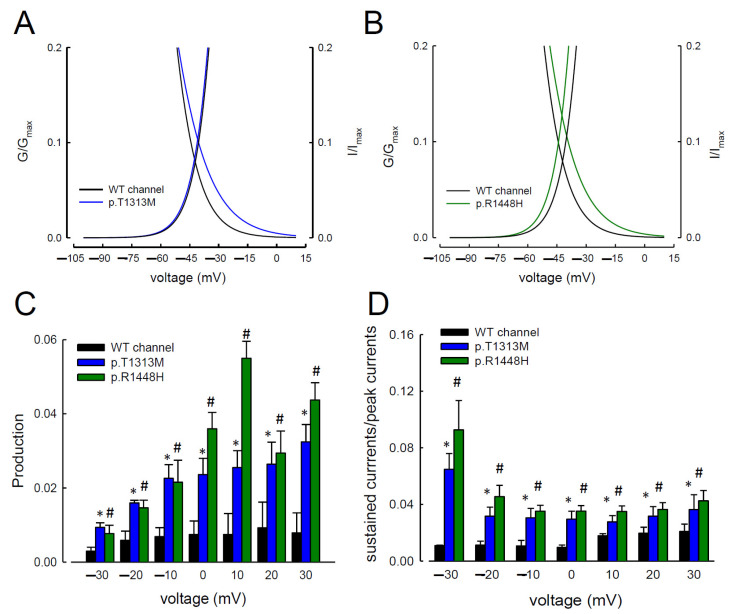
Extensive Window Currents in the p.T1313M and p.R1448H Mutant Channels. (**A**,**B**) The black (WT) and colored (p.T1313M, blue; p.R1448H, green) fitted lines represent a closer view of the activation and inactivation curves shown in Figure 1B between −100 and +10 mV in the WT, p.T1313M, and p.R1448H mutant Na_v_1.4 channels. (**C**) The production of G/G_max_ and I/I_max_ from Figure 2A,B are plotted against the specific voltage at −30 to +30 mV in the WT, p.T133M, and p.R1448H mutant Na_v_1.4 channels (* and #, *p* < 0.05). (**D**) The ratio between the sustained (the average currents between 85 and 90 ms of the pulse) and maximal (peak) transient Na^+^ currents at +30 to −30 mV (see Figure 1A for the sample sweeps) was significantly larger in the p.T1313M and p.R1448H channels than in the WT Na_v_1.4 channels (* and #, *p* < 0.05).

**Figure 3 biomedicines-09-00051-f003:**
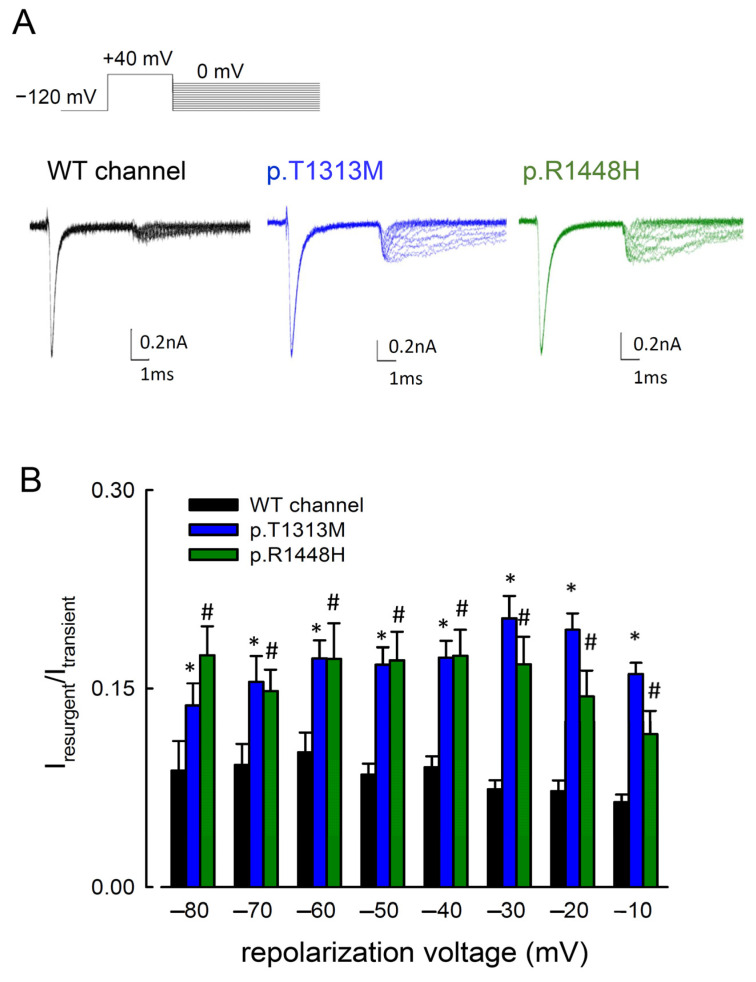
Resurgent Na^+^ Currents in the WT, p.T1313M, and p.R1448H Mutant Na_v_1.4 Channels. (**A**) Sample sweeps were obtained with 100-μM Na_v_β4 peptide for the WT, p.T1313M, and p.R1448H mutant Na_v_1.4 channels. The patch cells were held at −120 mV, and the resurgent Na^+^ currents of WT, p.T1313M, and p.R1448H mutant Na_v_1.4 channels were provoked by pulses between 0 and −120 mV in 10-mV increments following a depolarizing prepulse of +40 mV for ~10 ms. (**B**) Cumulative results were obtained from the experimental protocol described in Figure 3A for the WT, p.T1313M, and p.R1448H mutant Na_v_1.4 channels (*n* = 5). The ratio between the resurgent and peak transient currents (I_resurgent_/I_transient_) was significantly smaller in the WT channels than in the p.T1313M and p.R1448H channels at repolarization potentials between −80 and −10 mV (* and #, *p* < 0.05).

**Figure 4 biomedicines-09-00051-f004:**
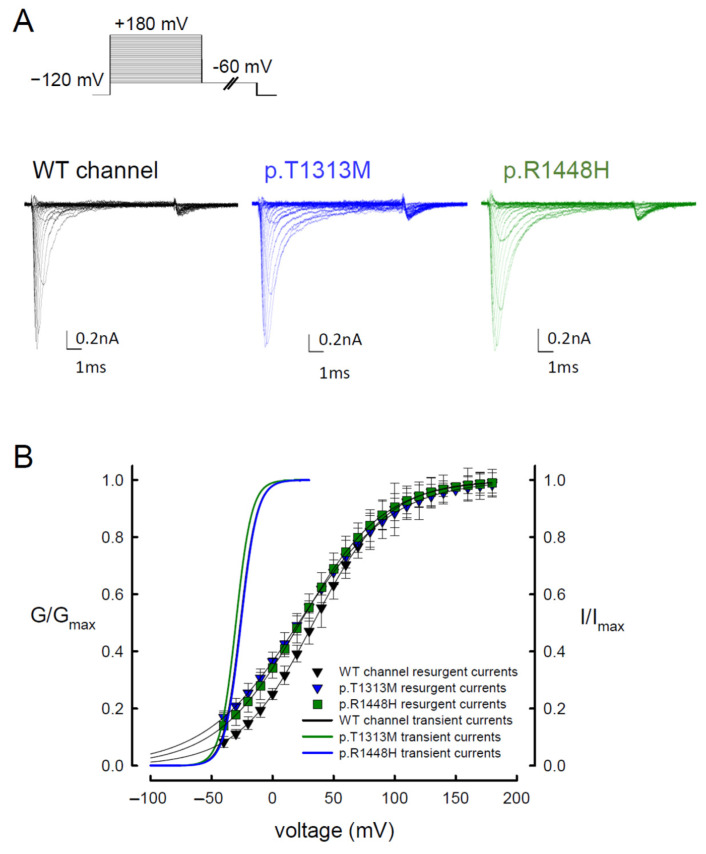
The Steady-state Activation Curves of Resurgent Na^+^ Currents in the WT, p.T1313M, and p.R1448H Mutant Na_v_1.4 Channels. (**A**) Sample sweeps were obtained with 100-μM Na_v_β4 peptide for the WT, p.T1313M, and p.R1448H mutant Na_v_1.4 channels. The cell was held at −120 mV for approximately 30 ms and then subjected to various depolarizing prepulses between −60 and +180 mV for ~5 ms in 10-mV increments. Resurgent Na^+^ currents were evoked by a repolarization pulse at −60 mV for ~150 ms. (**B**) The steady-state activation curves of the transient (the fitting lines in Figure 1B) and resurgent Na+ currents in the WT, p.T1313M, and p.R1448H mutant Na_v_1.4 channels are replotted for comparison. The resurgent Na^+^ activation curve of the WT Na_v_1.4 channel for each cell was obtained by fitting with Boltzmann functions, and the cumulative results for V_h_ and k were +33.5 ± 1.44 mV and 30.62 ± 1.1 for ~10-ms prepulses, respectively (*n* = 5). Those for p.T1313M mutant channels were +21.32 ± 3.11 mV and 38.59 ± 2.36 (*n* = 5; *p* < 0.05), and those for p.R1448H mutant channels were +22.56 ± 1.74 mV and 34.37 ± 1.35 (*n* = 5; *p* < 0.05).

**Figure 5 biomedicines-09-00051-f005:**
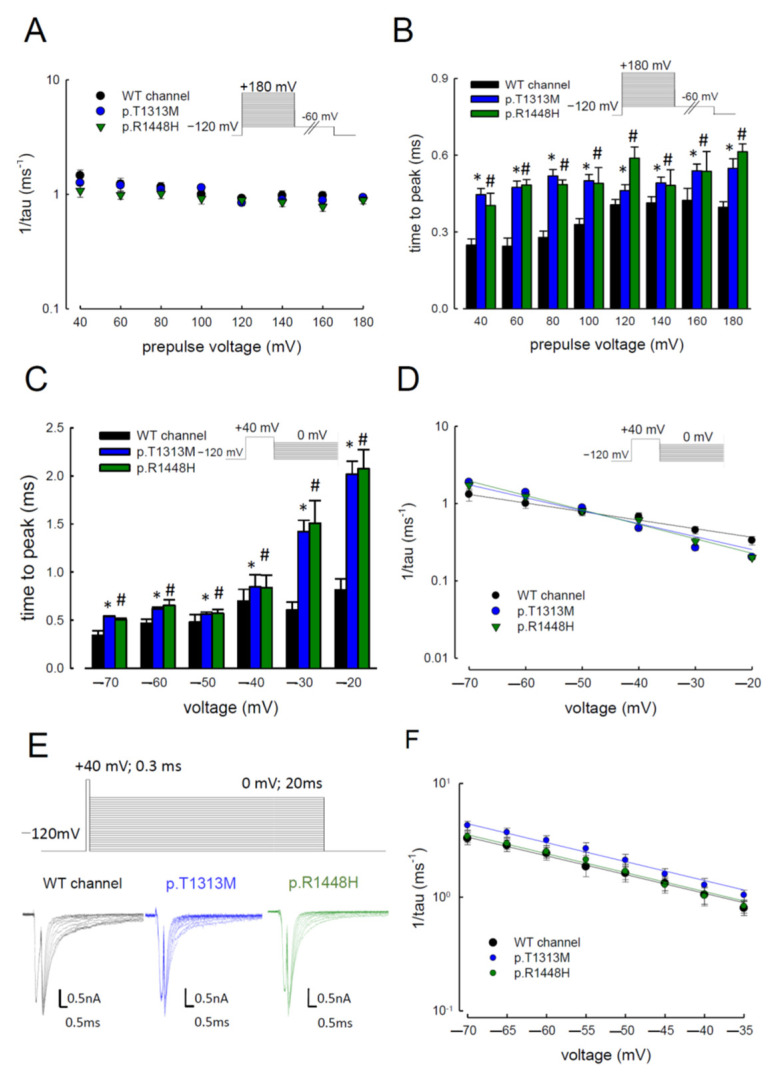
Unchanged Decay Kinetics but Slower Time to Peak for Resurgent Na^+^ Currents in the Mutant Channel. (**A**) The reverse decay time constants (1/tau) of the resurgent Na^+^ currents were obtained at the repolarization, −60 mV after the prepulses of +40 to +180 mV. The inverse decay time constants (1/tau) were comparable among the WT, p.T1313M, and p.R1448H Na_v_1.4 channels. (**B**) Cumulative results were obtained using the same experimental protocols in Figure 4A (*n* = 5). The time to peak for the resurgent Na^+^ current at different prepulse voltages in the p.T1313M and p.R1448H channels was consistently slower than that in the WT Na_v_1.4 channel (* and #; *p* < 0.05). (**C**) The time to the resurgent Na^+^ current peak was measured using the same protocol as used in Figure 3 and plotted against the repolarization potentials in the WT, p.T1313M, and p.R1448H mutant Na_v_1.4 channels. The time to peak for the resurgent Na^+^ current was significantly larger in the p.T1313M and p.R1448H channels than in the WT Na_v_1.4 channels, with repolarization membrane potentials between −20 and −70 mV (* and #; *p* < 0.05). (**D**) The reciprocal time constants (1/tau) for the decay phase of resurgent Na^+^ currents were plotted against the different repolarizing potentials in semi-logarithmic scales for the WT, p.T1313M, and p.R1448H mutant Na_v_1.4 channels. The lines were linear regression fits of the formula: 1/tau_(V)_ = 0.22 × exp(−0.64 V/25) ms^−1^, 0.12 × exp(−0.97 V/25) ms^−1^, and 0.09 × exp(−1.07 V/25) ms^−1^ for the WT, p.T1313M, and p.R1448H channels, respectively, where V is the membrane potential in mV. (**E**) Transfected cDNA CHO-K1 cells were held at −120 mV and stepped to +40 mV for ~0.3 ms of the activation pulse, followed by repolarization from −120 mV to 0 mV for ~20 ms in the WT, p.T1313M, and p.R1448H mutant Na_v_1.4 channels. The tail currents showed faster decay kinetics as the deactivating pulse became more negative. (**F**) The decay phase of tail currents in Figure 5E was fitted using mono-exponential functions for different deactivating potentials in the WT, p.T1313M, and p.R1448H mutant Na_v_1.4 channels. The inverse of time constants of the decaying phase in tail currents was plotted against voltage in semi-logarithmic scales for the WT, p.T1313M, and p.R1448H mutant Na_v_1.4 channels. The lines were linear regression fitted for the formula: 1/tau_(V)_ = 0.24 × exp(−0.95 V/25) ms^−1^, 0.29 × exp(−0.96 V/25) ms^−1^, and 0.24 × exp(−0.96 V/25) ms^−1^ for the WT, p.T1313M, and p.R1448H mutant channels, respectively, where V is the membrane potential in mV.

**Figure 6 biomedicines-09-00051-f006:**
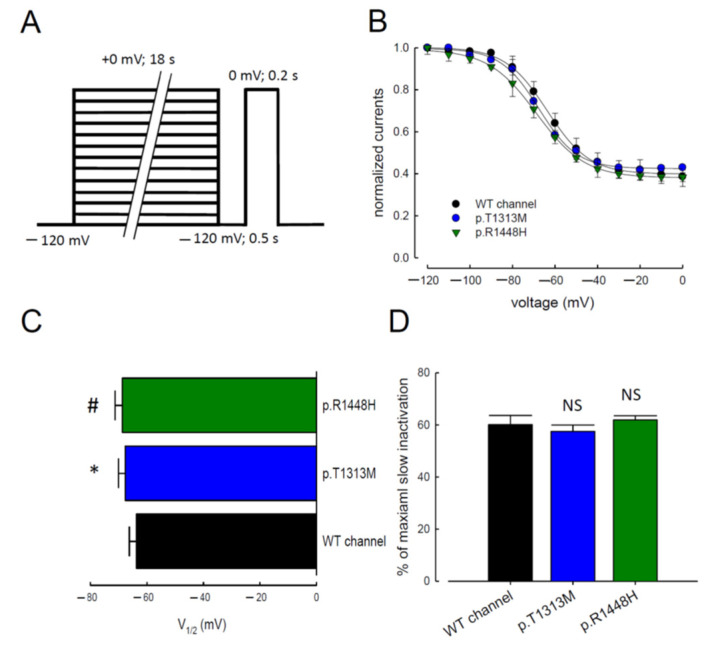
Slow Inactivation in the WT, p.T1313M, and p.R1448H Mutant Na_v_1.4 Channels. (**A**) The patched cells were held at −120 mV, and the pulse protocol was repeated every 30 s. Inactivating prepulses of different voltages (from −120 to 0 mV, ~18 s) were followed by a gap voltage at −120 mV for 0.5 s (to allow recovery from fast inactivation), after which the fraction of available channels was measured by a test pulse at 0 mV for 0.2 s. (**B**) The normalized peak Na^+^ currents were reported as a function of the conditioning pulse voltage. The WT, p.T1313M, and p.R1448H mutant Na_v_1.4 channels were fitted to slow inactivation curves using Boltzmann function, including a residual current of the formula: I_R_ + (1 − I_R_)/[1 + exp ((V − V_1/2_)/k)], where V is the membrane voltage. Since some channels do not enter slow inactivation, residual current (I_R_) was introduced into the equation to fit the voltage dependence of slow inactivation. V_1/2_ is the half-maximal voltage of slow inactivation and *k* is the slope factor. (**C**) The horizontal bar graph shows the V_1/2_ shift of the p.T1313M and p.R1448H channels than the WT Na_v_1.4 channel. Data are presented as the mean ± SEM (*n* = 5; * and #; *p* < 0.05). (**D**) The cumulative results indicated the peak percentage of the maximal slow inactivated channels, measured as [(1 − I_Residual currents_) × 100]. Data are presented as the mean ± SEM (*n* = 5; N.S., no statistically significant difference).

**Figure 7 biomedicines-09-00051-f007:**
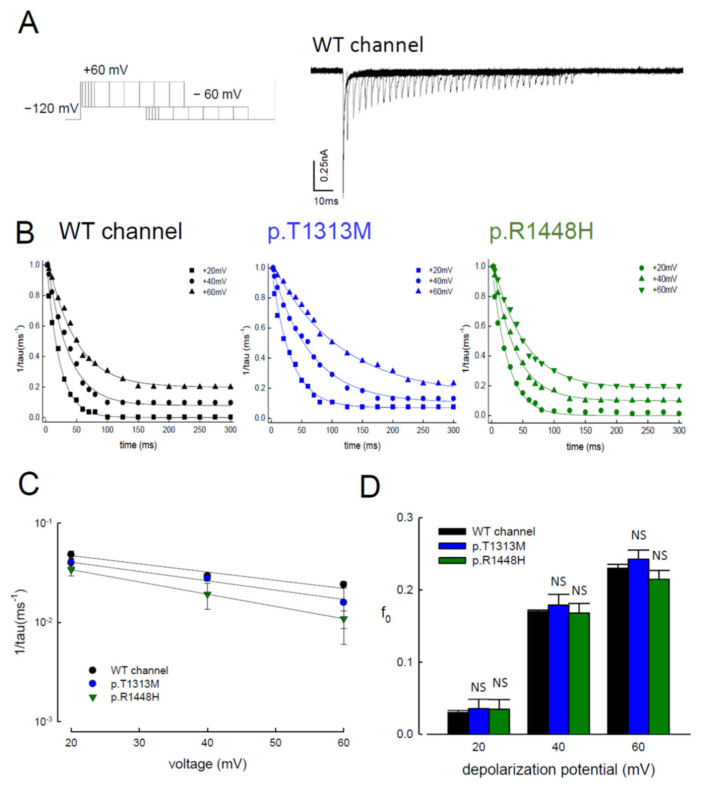
Resurgent Na^+^ Currents Decreased with Prolonged Depolarization Prepulses. (**A**) The cell was first held at −120 mV and then stepped to a gradually lengthened depolarization prepulse at +60 mV before stepping to −60 mV for ~300 ms to document the resurgent Na^+^ currents. The resurgent Na^+^ currents decreased with the lengthening of the prepulse in the WT Na_v_1.4 channels. (**B**) The normalized amplitude of the resurgent Na^+^ currents (normalized to the first current in each series) was plotted against the length of the prepulse (to +20, +40, and +60 mV, according to the protocol in Figure 7A). The lines were fitted to the data points using the following formula: normalized resurgent currents = (1 − f_0_) × exp [−(x − 4)/*τ*] + f_0_, where x is the prepulse length in ms for the WT, p.T1313M, and p.R1448H mutant Na_v_1.4 channels at different depolarization potentials. (**C**) The inverse of the time constant in Figure 7B was plotted against the prepulse voltage using a semi-logarithmic scale. The data were fitted with the following equation: 1/tau_(V)_ = 0.069 × exp(−0.5 V/25) ms^−1^, 0.069 × exp(−0.48 V/25) ms^−1^, and 0.064 × exp(−0.8 V/25) ms^−1^ for the WT, p.T1313M, and p.R1448H mutant Na_v_1.4 channels, respectively, where V is the prepulse potential in mV. (**D**) The f_0_ (the residual resurgent Na^+^ currents) in Figure 7B were plotted against different depolarization potentials in the WT, p.T1313M, and p.R1448H mutant Na_v_1.4 channels. There was no significant difference between the channels at each voltage (*n* = 5; N.S., no statistically significant difference).

**Figure 8 biomedicines-09-00051-f008:**
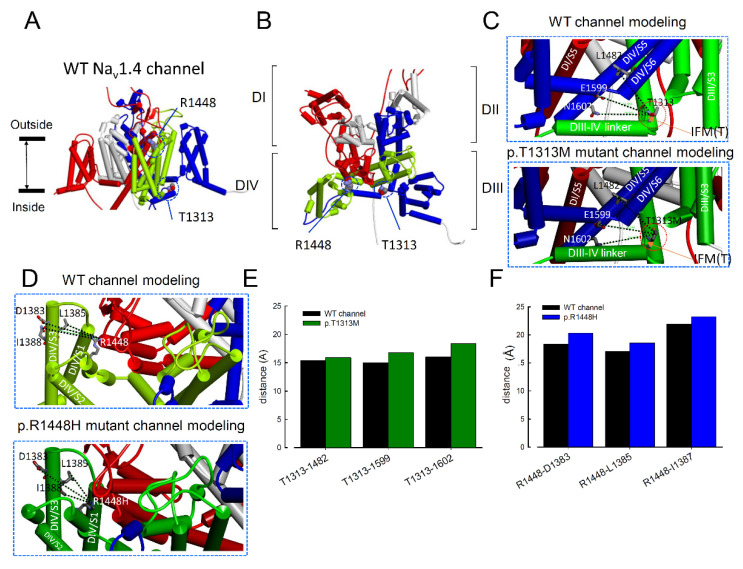
Homology Modeling of the Human WT and Mutant Na_v_1.4 Channels. (**A**) The homology model was constructed based on the correlative X-ray crystal structures of Na_v_1.4 using Discovery Studio 2018 [18,19,20]. Four subunits of WT Na_v_1.4 homology modeling were shown. The side view of the homology model of the WT Na_v_1.4 channel shows the transmembrane α-helix of the four domains. Domains I, II, III, and IV are colored red, white, blue, and green, respectively. The side chain of T1313M in the domain III–IV linker is indicated in the CPK model. The side chain of R1448H is located in the S4 of domain IV. (**B**) Regional view of the extracellular side of the pore in the homology model of the WT Na_v_1.4 channel. (**C**) A diagram of two domains (DIII and DIV) in the homology models of WT and p.T1313M mutant Na_v_1.4 channels. The side chains of p.T1313 to the p.L1482, p.F1599, and p.N1602 are indicated with sticks of different colors. The enlarged view of the boxed portion shows the inter-residue distances of ~15.3 Å, ~14.9 Å, and ~15.9 Å for p.T1313 to p.L1482, p.T1313 to p.F1599, and p.T1313 to p.N1602 in the WT channel, respectively. In the p.T1313M mutant channel, the distances were ~15.9 Å, ~16.7 Å, and ~18.4 Å, respectively. Arrowhead to identified the appropriate position of IMF. (**D**) Diagram of two domains from the homology models of the WT and p.R1448H Na_v_1.4 channels. The side chains of p.R1448 to the residues p.D1383, p.L1385, and p.I1387 are indicated with sticks of different colors. In the WT Na_v_1.4 channel, the inter-residue distances were ~18.32 Å, ~17.02 Å, and ~21.91 Å for p.R1448 to p.D1383, p.R1448 to p.E1385, and p.R1488 to p.N1602, respectively. In the p.R1488H mutant channel, they were ~20.32 Å, ~18.54 Å, and ~22.25 Å, respectively. (**E**) Summary plot to show the relative inter-residual distances (from tip-to-tip of the side chain) of p.T1313 (or p.T1313M) to L1482, p.T1313 to p.F1599, and p.T1313 to p.N1602 in the homology model. Distances between residues were increased in the p.T1313M mutant channel than in the WT Na_v_1.4 channels. (**F**) Summary plot of the relative distances (from tip-to-tip of the side chain) of residues p.R1448 (or p.R1448H) to p.D1383, to p.L1385, and to p.I1387 in the homology model. Increased distances between residues were observed in the p.R1448H mutant channel than in the WT Na_v_1.4 channel.

**Table 1 biomedicines-09-00051-t001:** Parameter of voltage-dependent activation and steady state fast inactivation of WT and mutant channels.

WT Channel				p.T1313M				p.R1448H			
Activation		Inactivation		Activation		Inactivation		Activation		Inactivation	
V_h_ (mV)	*k*	V_h_ (mV)	*k*	V_h_ (mV)	*k*	V_h_ (mV)	*k*	V_h_ (mV)	*k*	V_h_ (mV)	*k*
−24.7 ± 1.5	6.9 ± 0.3	−61.9 ± 0.9	−8.9 ± 0.3	−61.9 ± 0.9	−8.9 ± 0.3	−68.6 ± 0.3	−12.8 ± 0.2	−30.0 ± 1.5	6.4 ± 0.5	−64.78 ± 0.5	−11.6 ± 0.4

## Data Availability

Data is contained within the article or Appendix A.

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
