# Peer review of "Changes in Resurgent Sodium Current Contribute to the Hyperexcitability of Muscles in Patients with Paramyotonia Congenita"

_biomedicines, 2021, doi:10.3390/biomedicines9010051_

Round 1

Reviewer 1 Report

In the current manuscript, Huang and colleagues, investigated the possible roles of the two canonical NaV1.4 channel mutations, T1313M and R1448H in paramyotonia congenita (PMC) disorder. Using a synthetic NaV1.4 b4 peptide, the authors established that these mutations significantly enhance the resurgent currents of the NaV1.4 channels as well alter the biophysical characteristics of the wildtype NaV1.4 channels. The findings of this work help elucidate the mechanisms responsible for the hyperexcitability of the muscle membrane in the PMC disorder. Overall the manuscript is well written, and the data presented is of good quality. However, the manuscript requires some additional control experiments to demonstrate that some of the observed gating effects are due to the mutations but not due to the NaV1.4 b4 peptide. Also, the discussion is more generalised, and it should be extended to include the temperature dependence of the resurgent currents and the likely pathophysiological effects as a result of changes in the gating characteristics (please see comments 2 and 3 below).

Major comments:

  1. It makes sense to include the NaV4 b4 peptide in the pipette solutions to induce resurgent currents. However, I do not seem to understand why the authors measured the basic gating properties (activation, inactivation etc) and sustained currents of the mutant channels in the presence of this peptide (line 195)? How can one discard the possibility that the alterations in the normal gating properties of mutant channels is not due to the presence of this peptide? To understand the intrinsic properties of these mutant channels, experiments showing the gating effects of these three channels with no b4 peptide in the solutions should be included and compared against b4 peptide containing solutions. In order to visualise the effectiveness of b4 peptide, a current record with no resurgent currents (i.e. experiments with no peptide and same voltage protocol) should be added as a supplementary data.
  2. As mentioned by the authors several times in the text that the symptoms of PMC (e.g. muscle stiffness) are aggravated by cold temperatures. The authors performed their experiments at 25°C, which is a controlled room temperature. Can the larger resurgent currents account for the clinical symptoms of PMC and its exacerbation by cold? What could be the biophysical effects (inactivation, resurgent, persistent currents etc.) of these mutations at more reduced temperatures (e.g. 15°C)?  All of these need to be addressed in the discussion.
  3. To reproduce the physiological setting, the addition of full b4 subunits (by transfection) would provide the accurate information about the enhancement of resurgent currents by these mutations. However, as the authors already mentioned that the molecular basis for the resurgent currents is currently unknown. In this scenario, how one could argue that changes in resurgent currents are primarily responsible for the muscle hyperexcitability in the PMC? For instance, these mutations displayed an enhanced sustained current than the WT channels. These sustained currents could lead to a stable depolarization of the resting membrane potential which can potentially lead to hyperexcitability and subsequently paralysis. These correlations/contributions of each gating effect by the mutations to the clinical symptom of PMC should be discussed further.
  4. The topological changes shown by the homology modelling should be interpreted well and explained better in the corresponding sections. The current result section is over simplified and just quotes the distances with the surrounding residues (and nothing was mentioned in the discussion too about the modelling results). How the changes in distance of these mutants may influence the gating and genesis of the resurgent currents? In R1448H mutation, the positive charge arginine on domain IV is replaced by a neutral residue histidine. Does this alter the stabilization of the channel structure and interactions with neighbouring residues indicated by the model leading to fast inactivation etc.? Also, can the binding affinity of the NaV4 b4 peptide account for the increase in resurgent currents in the mutants?

Minor comments

  1. Is there any author missing in the manuscript or a typo? The author list ends with “and 6”.
  2. Can the authors highlight the region of the current traces where the sustained currents were measured? A box or similar showing this would be very helpful.
  3. It would be advantageous to the readers if the authors include a table with the parameters obtained from the Boltzmann fits in the WT and mutant channels.
  4. What does production exactly refer to in Fig 2: Panel C? I suppose these are the area under the curves from Panels A and B. The term “production” is very confusing since nothing new being produced here. Also, Generation is a better word for production in lines 179, 181, 207 etc.
  5. Greek alphabets (a, b) are not displayed in the pdf version of the manuscript.
  6. The authors are transfecting the CHO cells with the NaV1.4 plasmids for electrophysiology. Therefore, it is a transient expression but not stable expression. Please correct in lines 24, 169.
  7. The residues labelled in Fig. 8 (panels C, E) are extremely difficult to see. Please use a different way of labelling.
  8. Numerous typographical errors exist in the manuscript. Some examples are listed below.

         Line 74: window should be window currents

         Line 151: 2.6 Homology modelling is a sub-heading, but it is in section 2.5

         Line 222: DIVS4 should be DIV/S4

         Line 268: Tan should be than

Reviewer 2 Report

Review of “Changes in resurgent sodium current contribute to the hyperexcitability of muscles in patients with paramyotonia congenita, Huang C-W et al. Biomedicines

Overview:

This work examines the role of persistent and resurgent sodium current enhanced by two well known paramyotonia congenital mutations in SCN4A, the gene encoding the alpha subunit of the human skeletal muscle channel, hNaV1.4. Index patients with these mutations are described for criteria establishing paramyotonia congenita, followed by biophysical characterization of the mutations heterologously expressed in Chinese Hamster Ovary (CHO) cells. Gating parameters are provided for equilibrium (steady-state) activation and fast inactivation; a comparison of these shows the “window current” associated with a persistent (late) sodium current characteristic of myotonia mutations. Steady state slow inactivation is also compared between WT and mutant channels. The beta 4 peptide is used to generate resurgent currents whose properties and comparison of WT and mutation is the focus of the research presented. Mutations T1313M in the DIII-DIV linker and R1448H in the DIVS4 segment each enhance resurgent current and these currents exhibit distinct kinetics from tail currents shown. Homology models based on the cryo-EM structure of hNaV1.4 are created for WT and each mutant.

The experiments are robust and with a clear set of findings for resurgent current and the impact of the two PMC mutations on such current. A role for the effect of these mutations on that current in the pathogenesis of paramyotonia congenita and membrane hyperexcitability must be tempered with the fact that these findings are observed in a system employing a peptide to provoke resurgent current, and from a beta subunit not yet determined to play a physiological role in skeletal muscle. These results provide an additional set of biophysical characterizations of two well known PMC mutations and the novel gating defects may be shown to be an important determinant of patient phenotype once such determinations are made.

Major Points

Results section

3.5 Slow inactivation. There are two issues with the experiments to determine the relative effects of T1313M and R1448H on slow inactivation compared to wild type channels. First, the time of depolarization at 18 seconds is short, with measurements of onset of slow inactivation in NaV1.4 expressed in mammalian cells typically twice that or more. Equilibrium may not have been reached. Second, and related, is the time of hyperpolarization to recover from fast inactivation, here used as 500 ms.  This is a long hyperpolarzing conditioning segment, and is well within the time of substantial recovery from slow inactivation.  The results show that slow inactivation culminates with an asymptote of 40%, which is much closer to that observed in cardiac, NaV1.5 channels, than in skeletal muscle, NaV1.4 channels. The most likely reason for this discrepancy is the likelihood that slow inactivation was not fully achieved by the initial depolarization and that recovery of a significant number of channels from slow inactivation was also incurred in this protocol.

3.7 Homology modeling.  I do not see what conclusions can be gained from these models. There are measurements of distance, implying that specific amino acid interactions with R1448 and T1313 are disrupted by histidine and methionine substitutions, respectively. The amino acids for which distance is increased by these substitutions are given, but no mention of the reported role for any of these potential targets is given. One would hypothesize that substitution of T1313 might alter its interaction with putative receptor sites for the inactivation particle, and that substitution of R1448 might alter its interaction with countercharges in S1-S3 as shown by molecular dynamics simulation. The interactions proposed are not mentioned in this light here or in the discussion. Substitution of other residues would most likely also result in alteration of interatomic distances in a homology model. What is the significance of these increased distances?

Discussion and Interpretation

The impact of beta 4 on channel function has been established from the perspective that the beta 4 peptide can induce resurgent currents. In some sodium channels such a resurgent current appears to be an important component of the channel function and the generation of repetitive action potentials in neurons. Physiologically, this subunit may be involved in the regulation of cardiac or skeletal muscle sodium channels, but this remains to be determined based on studies that conclusively localize that subunit with alpha subunits in cardiac and skeletal muscle. The introduction of this peptide in order to study resurgent currents necessitates a limitation of the interpretation of the pathophysiology of T1313M or R1448H mediated paramyotonia congenita. At one point in the results, and in the discussion, there is a decided conclusion that resurgent currents are a determinant of hyperexcitability, and this conclusion is not supported, yet, by available evidence that hNaV1.4 mutations in patients would exhibit such resurgent currents to promote hyperexcitability.

Approach

Several approaches could be used to strengthen the notion that resurgent currents promote hyperexcitability. Mathematic models of skeletal muscle fibers and t-tubules have been used in action potential simulations to demonstrate the impact of specific mutations for sodium channel myotonia, paramyotonia congenita, hyperkalemic and hypokalemic mutations on excitability. For instance, persistent current in hyperkalemic periodic paralysis, in such models, promoted membrane depolarization and paralysis.  Does the impact of T1313M and R1448H mutation on resurgent current promote hyperexcitability in such a model (by itself); does it contribute to hyperexcitability along with other gating defects?  Together with the lack of complete understanding of the native role of beta 4 subunit in skeletal muscle fiber physiology, the results should either be extended in such a model, or the interpretation attenuated at present.

Minor points

Introduction

Line 55: The well conserved IFM(T) motif is the established inactivation particle for sodium channels. Given that mutation T1313M is part of or adjacent to this motif, this structural determinant in the DIII-IV linker needs to be identified.

Next paragraph: Slowed entry into fast inactivation and accelerated recovery from fast inactivation are well established, gain of function effects of PMC mutations that are not mentioned here.

Experimental Section

2.1. Patients and Genetic Analysis. The methods for clinical assessment of patient phenotype need to be given here, instead of the results from such assessment. Genetic analysis of what tissue?

2.5, line 154: The structural data for hNaV1.4 needs the reference to that work (Pan et al., 2018) and this data is cryo-EM not X-ray crystallography.

2.7. Data Analysis: “data” are plural and as such Student’s independent t-test(s) are appropriate grammar.

Results

3.1 . There should be a separate section on the clinical assessment. More detail on these index patients is warranted. For the variants, this section reads as both index patients harbor both mutations. If not, state, If so or if one patient harbors both mutations, then biophysical characterization of the effects of each mutation should be supplemented with such for both mutations expressed concurrently.

Line 176 and Figures 1, 2:  “The predicted window current or the presumed sustained Na+ current was defined as the measurement of area under the activation and inactivation curves.”  Is this measurement the “production of G/Gmax and I/Imax” given in the legend for Figure 2? There needs to be a clear explanation of the means by which window current is quantified.

First, if “production of G/Gmax and I/Imax” is the measurement of area of window current, how is this measured?  Is this the “product” of G/Gmax and I/Imax then” If so how does that equate to area under the curve?

Second, Figure 2D is generated from the experiments shown in Figure 1 and as a panel belongs in Figure 1. If the point is to correlate area under the curve to sustained current, make that comparison of this relationship over a voltage range, as panel 2D.

3.2. line 223: “The resurgent Na+ current plays a key role in the transition states between the inactive and open states of the NaV1.4 channels”. First, this statement is not a result but an interpretation. It is based on experiments utilizing a beta 4 peptide to induce resurgent current, and for which the presumptive beta 1 subunit in skeletal muscle is not demonstrated to participate in channel gating in this report.  This is a larger question, see Major Points, but for the purposes of this section it is suggested to avoid interpretations about the physiological role for a phenomenon that is observed only for cells treated with a peptide whose role in skeletal muscle is not yet determined equivocably. Other studies have shown that this sodium channel does not recover through the open state but through closure.

3.3. line 244: “According to the proposed pore block model, the competition……should favor the leftward shift of the resurgent Na+ current activation curve (hyperpolarization), which was disparate to our finding”. First, there is assumption here that the beta 4 peptide and the inactivation particle compete for block of the pore. The general consensus is that the inactivation particle binds to a receptor and this interaction then promotes inactivation. The location of this receptor is still under investigation with sites in DIII S4-S5 linker, DIV S4-S5 linker, and S5 segments as putative sites. At the very least there is no general consensus that the inactivation particle blocks an open pore. Second, the inactivation particle is not 50 amino acids in length. The DIII-DIV linker is, but the inactivation particle is a tripeptide IFM motif (or extended by some as an IFMT motif). Thus, the inactivation particle is, itself, shorter in length than the beta 4 peptide. Regardless, this particle is also part of the channel and resides in a relatively structured aspect of the DIII-DIV linker; it is not a peptide. Finally, a comparison of competitiveness to the pore or a receptor between a soluble peptide, to this intrinsic portion of the channel, based on size assumes that the particle is tethered but without any restriction excepting its size, which is not accurate.

Other:  It might be helpful to compare more directly earlier work on these mutations and gating parameters, with respect to same as those studied here, or other, from the references provided. There are additional biophysical reports on T1313M and R1448H for gating parameters and a report on the effect of R1448C to enhance ramp currents.

Round 2

Reviewer 1 Report

As requested, the authors have provided the necessary information for doing all the experiments in the presence of Nav1.4 B4 peptide. The text/figures/tables have also been modified following the suggestions in order to facilitate the reading and to avoid confusion. The authors have adequately addressed all of my previous criticisms/comments and I have no additional concerns.